# Developmental Ethanol Exposure Impacts Purkinje Cells but Not Microglia in the Young Adult Cerebellum

**DOI:** 10.3390/cells13050386

**Published:** 2024-02-23

**Authors:** MaKenna Y. Cealie, James C. Douglas, Hannah K. Swan, Erik D. Vonkaenel, Matthew N. McCall, Paul D. Drew, Ania K. Majewska

**Affiliations:** 1Department of Neuroscience, School of Medicine and Dentistry, University of Rochester, Rochester, NY 14620, USA; makenna_cealie@urmc.rochester.edu; 2Department of Neurobiology and Developmental Sciences, University of Arkansas for Medical Sciences, Little Rock, AR 72205, USA; jdouglas@uams.edu (J.C.D.); drewpauld@uams.edu (P.D.D.); 3Department of Biostatistics and Computational Biology, School of Medicine and Dentistry, University of Rochester, Rochester, NY 14620, USA; hannah_swan@urmc.rochester.edu (H.K.S.); matthew_mccall@urmc.rochester.edu (M.N.M.); 4Earth and Biological Systems Division, Pacific Northwest National Laboratory, Richland, WA 99354, USA; erik.vonkaenel@pnnl.gov

**Keywords:** microglia, cerebellum, Purkinje cell, immune system, ethanol, fetal alcohol spectrum disorders (FASD), prenatal alcohol exposure, neurodevelopment, two-photon imaging

## Abstract

Fetal alcohol spectrum disorders (FASD) caused by developmental ethanol exposure lead to cerebellar impairments, including motor problems, decreased cerebellar weight, and cell death. Alterations in the sole output of the cerebellar cortex, Purkinje cells, and central nervous system immune cells, microglia, have been reported in animal models of FASD. To determine how developmental ethanol exposure affects adult cerebellar microglia and Purkinje cells, we used a human third-trimester binge exposure model in which mice received ethanol or saline from postnatal (P) days 4–9. In adolescence, cerebellar cranial windows were implanted and mice were aged to young adulthood for examination of microglia and Purkinje cells in vivo with two-photon imaging or in fixed tissue. Ethanol had no effect on microglia density, morphology, dynamics, or injury response. However, Purkinje cell linear frequency was reduced by ethanol. Microglia–Purkinje cell interactions in the Purkinje Cell Layer were altered in females compared to males. Overall, developmental ethanol exposure had few effects on cerebellar microglia in young adulthood and Purkinje cells appeared to be more susceptible to its effects.

## 1. Introduction

Fetal alcohol spectrum disorders (FASD) are a group of conditions caused by prenatal alcohol exposure characterized by cognitive, behavioral, and physical impairments [1,2]. In humans, FASD are linked with cerebellar impairments which manifest as behavioral deficits in movement and coordination [3]. Indeed, decreases in cerebellar size have been reported in individuals with FASD and have been replicated in animal models where they are accompanied by ethanol-induced cell death [4,5]. The cerebellum may be particularly vulnerable to ethanol exposure due to its unique developmental timeline. Unlike the cortex, where neurogenesis and migration conclude by birth, the cerebellum undergoes postnatal development with neurogenesis and migration continuing after birth in humans and mice [6]. Because of this protracted development, ethanol exposure late in pregnancy, such as during the third trimester, which is the peak of neurogenesis in the human cerebellum (and is equivalent to the early postnatal period in mice), might have a disproportionate effect on cerebellar development, causing cerebellar cell death and altering cerebellar developmental trajectories [6]. These acute changes due to ethanol may have long-lasting effects that persist into adulthood. In humans, lower cerebellar volumes were found in both adolescents and young adults with FASD compared to controls [7,8], and motor deficits in adults with FASD have also been reported [9]. These prolonged deficits may be due to the loss of cerebellar neurons, as lower densities of these have been reported in animal models in adulthood [10], or due to changes in neuronal circuit function, glial activity, or interactions between neurons and glia [11]. 

Ample evidence from animal models suggests that Purkinje cells, the sole output of the cerebellar cortex, may be particularly vulnerable to developmental ethanol exposure. Ethanol exposure during the third-trimester equivalent in mice (postnatal days (P) 4–9) has been found to lead to Purkinje cell death, with fewer cells found after exposure both in development and later in life [4,10,12,13]. During this early postnatal time, Purkinje cells are migrating and forming a monolayer in the Purkinje Cell Layer (PCL), and therefore may be particularly vulnerable to ethanol exposure [14,15]. Within the first three postnatal weeks, rodent Purkinje cell arbors grow and spread their dendrites in the Molecular Layer (ML) [16,17]. This first postnatal month in mice is also critical for cerebellar circuit development as various other cells are migrating, maturing, and forming synapses on Purkinje cell somas and dendrites [18]. Alterations in cerebellar circuitry after developmental ethanol exposure have been found, including decreased Purkinje cell excitability, increased firing, and altered synaptic interactions in rodents [19,20]. Aberrations in this early postnatal developmental period may underlie the deficits seen in adults subject to developmental ethanol exposure. 

Another cell type developing during the early postnatal period is microglia, the immune cells of the central nervous system. During the first postnatal week, microglia migrate from the white matter into the grey matter of the cerebellum, and then colonize the PCL and ML throughout the first postnatal month [21]. Like Purkinje cells, cerebellar microglial cell death and reduced numbers have been reported when rodents are exposed to ethanol during development [22]. Cerebellar microglia that survive developmental ethanol exposure alter their phenotype to one that suggests reactivity, with altered microglia morphology and release of inflammatory factors [4,23]. Microglia may impact neuronal development through these inflammatory actions, or through the loss of their normal roles in the homeostatic brain. Microglia are highly dynamic cells that survey their surroundings and interact with neurons influencing synaptic pruning and formation [21,24,25,26,27]. Microglia have been found to interact in vivo with Purkinje cells in the cerebellum of the healthy adult brain [28], providing a role for these interactions to alter Purkinje cell activity in the adult.

We previously examined how microglia dynamics and their interactions with Purkinje cells were affected in adolescent mice that were given ethanol from P4–9 [24], the equivalent of the third trimester of human development, when the cerebellum may be the most susceptible to ethanol. Ethanol elicited only subtle changes in Purkinje cells, microglia, and the interaction of the two cell types, with trends towards increased ML microglia–Purkinje cell interactions in ethanol-dosed compared to saline-dosed animals [24]. However, this work made us question whether new effects on these cell types emerge as animals mature. In fact, much of the work examining cerebellar microglia and Purkinje cells has examined the period immediately after ethanol exposure, instead of later in life, and many of these studies are limited to fixed tissue. Additionally, these two cell types have largely been studied individually and their interactions have only recently been examined. Therefore, our current study seeks to address these gaps and answer the question of whether developmental ethanol exposure has long-lasting impacts on cerebellar microglia, Purkinje cells, and their interactions in vivo as well as in fixed tissue. We gave mice a binge-level dose of ethanol from P4–9 and then examined lobule IV/V of the cerebellar vermis in vivo using two-photon microscopy and in fixed sections using confocal imaging at ~P60. We found that ethanol did not affect in-vivo microglia dynamics, morphology, or injury response. Microglia density in fixed tissue was also largely unaffected. However, Purkinje cell linear frequency was significantly decreased by ethanol, particularly in females. Microglia–Purkinje cell interactions were affected by sex, with different effects in females compared to males in vivo and in fixed tissue. This work suggests that cerebellar microglia and their interactions with Purkinje cells are largely unaffected later in life after ethanol exposure during the human third-trimester equivalent, but supports evidence that Purkinje cells are particularly vulnerable to developmental ethanol exposure. 

## 2. Materials and Methods

### 2.1. Animals

The University of Rochester Committee on Animal Resources and National Institutes of Health Guidelines were strictly followed for all experimental protocols. Transgenic mice (N = 30) were bred in house on a C57BL/6 J background (Jackson Labs, Bar Harbor, ME, USA, strain 000664; RRID:IMSR_JAX:000664), were provided ad libitum chow and water, and exposed to 12 h of light and 12 h of darkness (6 AM lights on). To allow for the fluorescent visualization of microglia and Purkinje cells in the same mouse, L7^cre^/Ai9^+/−^/Cx3cr1^G/+^ mice were bred as described in [24]. In brief, GFP insertion into the *Cx3Cr1* locus allows the expression of GFP in microglia under control of the Cx3Cr1 promoter with a loss of one copy of *Cx3Cr1* in heterozygous Cx3cr1-eGFP mice (Jackson Labs strain 005582; RRID:IMSR_JAX:005582 [29]). Homozygous Cx3cr1-eGFP mice were bred to tdTomato reporter (Ai9; Jackson Labs strain 007909; RRID:IMSR_JAX:007909) mice to create Ai9^+/+^/Cx3cr1^G/G^ mice. The fluorescent labeling of Purkinje cells was achieved with the breeding of Ai9^+/+^/Cx3cr1^G/G^ mice to a Purkinje cell-specific cre line (L7-cre; Jackson Labs strain 004146; RRID:IMSR_JAX:004146) to create the final experimental L7^cre^/Ai9^+/−^/Cx3cr1^G/+^ mice. Nineteen mice used in [24] were aged and also used in this study, as well as eleven additional mice not previously used experimentally (N = 30). To examine potential sex differences, both male and female mice were used and analyzed in all experiments. Postnatal day (P) 0 was considered the day of birth. 

### 2.2. Ethanol Dosing

Mice were subcutaneously dosed with either ethanol or saline as described in [24,30,31]. In brief, on P4, mice were toe-clipped for identification, and from P4 to P9, male and female mice were weighed and injected twice a day, 2 h apart, with either 5.0 g/kg/day ethanol solution (20% *v*/*v* ethanol in saline) or saline (0.9% NaCl). All pups were removed from the dam during dosing and placed on a heating pad to avoid potential hypothermia. We previously found that in L7^cre^/Ai9^+/−^/Cx3cr1^G/+^ mice, this dosage method produced blood ethanol concentrations (BECs) of ~450 mg/dL, a binge-level amount, 90 min after the last dose was delivered on both the first day of dosing (P4) and the last day of dosing (P9) [24]. Pups were returned to their dams in the breeding cages after injection and were weaned on P21 into same-sex cages. Although some pups briefly appeared lethargic following ethanol treatment, maternal behavior including nursing did not appear overtly different for ethanol- and saline-treated pups. Pups were not culled and litter size varied between 4 and 11 pups. Mixed litters were used with a maximum of four mice used per litter (no more than one male and one female taken for each of the ethanol and saline treatments from the same litter). Each analysis included n = 5 mice per sex per treatment. As some animals did not have sufficient window quality for in-vivo imaging, but could be used for fixed-tissue collection, thirty total animals were used, including ethanol, n = 14, eight males and six females; saline, n = 16, nine males and seven females, spread across 17 separate litters. Of the 19 mice used in [24], the treatment distribution was ethanol, n = 8, three males and five females; saline, n = 11, six males and five females. Of the 11 mice solely generated for this study, the treatment distribution was ethanol, n = 6, five males and one female; saline, n = 5, three males and two females. Data analysis was performed blind to exposure group and sex. 

### 2.3. Quantitative Real-Time PCR Analysis

A separate cohort of L7^cre^/Ai9^+/−^/Cx3cr1^G/+^ mice was used for real-time PCR (rtPCR) at P10. Thirty-eight pups were used, including ethanol, n = 19, nine males and ten females; saline, n = 19, ten males and nine females. Pups were spread across five litters, with multiple pups of the same treatment and sex being used from the same litter. The same P4–P9 dosing paradigm was used as for the rest of the mice in this study. For this cohort, on P10 (24 h after the second dose of ethanol on P9), pups were removed from their dams, euthanized with an overdose of sodium pentobarbital (Euthasol, Virbac, Westlake, TX, USA), and transcardially perfused with PBS containing 5 units/mL of heparin [23]. The brain was then removed, and the cerebellum was microdissected and submerged in RNAlater Stabilization Solution (Invitrogen, Waltham, MA, USA, Cat#AM7024), and stored at −20 °C. 

The cerebellum was then processed as in [23]. In brief, a PowerLyzer 24 homogenizer (Qiagen, Germantown, MD, USA) with 0.5 mm glass beads was used to homogenize the cerebellum. An miRNeasy Mini kit was used to isolate RNA, and DNA was removed with DNaseI on-column DNA digestion (Qiagen #217084 and #79254). Total RNA concentration was measured with a NanoDrop 2000 spectrophotometer (ThermoFisher Scientific, Wilmington, DE, USA; RRID:SCR_018042) and the iScript™ system (Bio-Rad, Hercules, CA, USA; # 1708891) was used to prepare cDNA from the mRNA template. Upon synthesis completion, the cDNA was diluted to 25 ng/μL with nuclease-free water and stored at 4 °C.

Real-time PCR was used to quantify mRNA levels with a Bio-Rad CFX Opus 96 Real-time PCR Detection system and TaqMan^®^ Gene Expression Assays (ThermoFisher #4331182) as in [23]. rtPCR was performed in duplicate with SsoAdvanced Universal Probes Supermix (Bio-Rad #1725285) and TaqMan FAM-MBG Primer/Probes. Baseline threshold CT values were obtained for each reaction and expressed as mean ΔCT relative to mean, duplicate β-actin control reactions for each sample. Gene expression fold differences between experimental groups were calculated using the ΔΔCT method. One pup (female ethanol) had particularly high transcript levels and was identified as a statistical outlier (ROUT, Q = 1%) for IL-1β and CCL2, but not TNF-α, and was thus excluded from subsequent analyses for all three molecules.

### 2.4. Cranial Window Surgeries 

As in [24], L7^cre^/Ai9^+/−^/Cx3cr1^G/+^ mice underwent cranial window surgery in adolescence between P26 and P31 (hereafter referred to as P28) to allow for in-vivo two-photon imaging of both microglia and Purkinje cells. Using methods previously described [24,28,30,31,32], mice were weighed and anesthetized with intraperitoneal (i.p.) injections of fentanyl cocktail (fentanyl 0.05 mg/kg; midazolam, 5.0 mg/kg; dexometomidine, 0.05 mg/kg). Aseptic technique was followed to surgically implant cranial windows over cerebellar lobules IV/V. Throughout all surgical procedures and later two-photon imaging, lubricant ointment was applied to the animal’s eyes and 37 °C body temperature was maintained. Partial scalp removal was followed by membrane clearance over the cerebellum. A glass coverslip replaced the skull over a 3 mm circular craniotomy. C&B Metabond dental cement (Parkell, Edgewood, NY, USA) was used to secure a headpost to the scalp and cover the rest of the exposed skull. After surgery, mice were given slow-release buprenorphine (5 mg/kg) for analgesia and monitored for 72 h. Some animals, such as those used in [24], were imaged using two-photon microscopy for up to ~one hour before being returned to their cages to recover. Other animals, including some that were added for the purposes of this study, were not imaged directly following surgery.

### 2.5. Two-Photon In-Vivo Imaging 

In young adulthood, between P52 and P64 (hereafter referred to as P60), L7^cre^/Ai9^+/−^/Cx3cr1^G/+^ mice were anesthetized (i.p.) with fentanyl cocktail (as described above) and affixed to an imaging stage. Two-photon in-vivo imaging was performed to capture microglia dynamics, morphology, and interactions with Purkinje cells. As in [24], a custom two-photon laser-scanning microscope (Ti: Sapphire, Mai-Tai, Spectra Physics, Milpitas, CA, USA; modified Fluoview confocal scan head, 20× water-immersion objective, 0.95 numerical aperture (NA), Olympus, Tokyo, Japan) achieved excitation with 100 fs laser pulses (80 MHz) at 920 nm with a power of ~30 mW measured at the sample. To simultaneously image microglia and Purkinje cells, a 565 dichroic with 520/40 (GFP, microglia) and 598/30 (Ai9, Purkinje cells) filters were used. For each animal, time-lapse imaging was performed over an hour at 5 min intervals to image 101 μm z-stacks at a 1 μm z-step size with an 800 × 600 pixel frame size with a 4× digital zoom, capturing both the Molecular Layer (ML) and Purkinje Cell Layer (PCL) as determined by the presence of the Purkinje cell dendrites or somas, respectively. 

To examine the microglial injury response, a laser ablation was inflicted in the parenchyma. Two different imaging systems were used to carry out this experiment. The custom system detailed above created a laser ablation by performing a point scan at 800 nm with a power of ~2.5 W in the ML. After the injury, time-lapse imaging at 920 nm with a power of ~30 mW resumed to collect 58 μm z-stacks at a 1 μm z-step size with an 800 × 600 pixel frame size with a 4× digital zoom every 5 min for an hour in the ML. The URMC Center for Advanced Light Microscopy and Nanoscopy (CALMN; RRID:SCR_023177) Olympus Fluoview FVMPE-RS two-photon microscope, equipped with two Ti:Sapphire lasers (InSightX3 and MaiTai HP DeepSee, Spectra Physics) and a 25× water immersion objective (XL Plan N 1.05 NA, Olympus), was also used for laser ablations. On this system, laser ablations were created with a point scan at 800 nm with a power of ~2.5 W. Then, time-lapse imaging at 920 nm resumed to collect 58 μm z-stacks at a 1 μm z-step size with an 800 × 600 pixel frame size with a 0.25 μm/pixel size (2.88× scan zoom to achieve the same pixel size as the custom 2p) every 5 min for an hour. A 570 dichroic with 495–540 (GFP) and 575–630 (Ai9) filters was used to image microglia and Purkinje cells, respectively.

In-vivo image analysis was performed in Ilastik ([33]; version 1.4.0rc6; RRID:SCR_015246), NIH ImageJ (imagej.net; version 1.54f; RRID:SCR_003070) or FIJI (imagej.net; version 1.54f; RRID:SCR_002285), and MATLAB (MathWorks, Natick, MA, USA; version R2020a; RRID:SCR_001622). Spectral overlap between the GFP microglia and Ai9 Purkinje channels was corrected in ImageJ/FIJI as in [24]. 

### 2.6. In-Vivo Image Analysis

Image analyses including image preprocessing, object classification, microglia dynamics, microglia Sholl analysis, and microglia–Purkinje cell interactions were performed as described in [24] and are described in detail below. 

#### 2.6.1. Image Preprocessing

A custom MATLAB script was used to blind all images after collection and prior to analysis. After blinding, images were drift-corrected in ImageJ/FIJI. As Purkinje cells could be seen in the microglia channel after imaging, ImageJ/FIJI was used to subtract the Ai9 Purkinje cell channel from the GFP microglia channel [31]. The two channels were separated and further split into 12 separate time points. Each time point was corrected for background fluorescence. The amount of bleed-through from the Ai9 channel into the GFP channel was calculated and the appropriate correction applied to the Ai9 channel. Then the brightness-compensated Ai9 Purkinje cell channel was subtracted from the GFP microglia channel. Microglia and Purkinje cells channels then underwent principal component analysis (PCA) with a custom MATLAB script to remove image noise. Substacks dividing the ML and PCL of the cerebellum were created in ImageJ/FIJI. Layer separation was determined through the manual visual identification of Purkinje cell somas, considered the start of the PCL and the end of the ML. A consistent number of z slices was chosen for each substack. 

#### 2.6.2. Object Classification

Ilastik, an image classification and segmentation software, was used to detect microglia and Purkinje cells automatically. Training was performed by the manual tracing of microglia in both the ML and PCL for either processes or somas, as well as Purkinje cell dendrites or branch points in the ML and Purkinje cell somas in the PCL. During object classification, appropriate thresholding and size exclusion criteria were chosen. The object output of microglia and Purkinje cells was then binarized. Finally, whole microglia and whole Purkinje cell images were created in ImageJ/FIJI by adding microglia processes and somas, as well as Purkinje cell dendrites and branch points. 

#### 2.6.3. Microglia Dynamics

For both the ML and PCL, microglia motility and surveillance were analyzed. ImageJ/FIJI was used to max-project and drift-correct each of the 12 time points for both the ML and PCL leading to 24 max z-projected images for each animal [34,35]. Cells were identified in Ilastik as described above. Microglia motility and surveillance were determined with a custom MATLAB script by comparing the 12 time points. The sum of the thresholded extended and retracted pixels was divided by stable pixels between consecutive time points to calculate the motility index (M.I.) of the microglial processes to assess how microglia sample their environment. The number of microglia pixels in the maximum projection of all time points was normalized to the number of microglia pixels in the first time point to calculate the surveillance ratio (S.R.) and thus examine how much of the parenchyma microglia survey over time given their initial morphology [28].

#### 2.6.4. Microglia Morphology: Sholl

Microglia arbor complexity was examined in the first time point with Sholl analysis. For both the ML and PCL separately, one microglia was selected that was fully contained in the max-projected images (projection described in microglia dynamics) and its processes were manually traced in ImageJ/FIJI. The Sholl analysis plug-in was run in ImageJ/FIJI with concentric rings at 2 μm intervals out to 70 μm from the soma center. The number of intersections across each ring was used to create Sholl curves.

#### 2.6.5. Microglia Laser Ablation

Microglia injury response to a laser ablation was measured in the ML with FIJI/ImageJ with a modified protocol from [28]. ImageJ/FIJI was used to drift-correct and max-project 30 μm in the z direction around the injury core for each of the 12 time points. In the first time point, the ImageJ/FIJI polygon tool was used to draw an ROI around the injury core. In each of the 12 time points, the polygon tool was used to draw an ROI around the “microglia front,” the border of the most proximal tips of the microglia processes. The areas of the core and each microglia front were measured. For each time point, the core area was subtracted from the microglia front area of that time point, then normalized to the front–core area of the first time point to determine microglia convergence on the injury core. 

#### 2.6.6. Microglia–Purkinje Cell Interactions

Microglia–Purkinje cells dynamic interactions were assessed over time in 3D. Cell identification in Ilastik was performed as described above for both ML and PCL non-max-projected z-stacks for each of the 12 time points. Whole-cell and individual components were quantified for volume fraction (number of pixels divided by the image volume) in ImageJ/FIJI. The overlap between microglia and Purkinje cells was determined by multiplying the binarized images. The pixel numbers of the resulting interactions were measured, normalized to the number of microglia pixels, and averaged across the 12 time points before being compared across conditions and sexes. 

The microglia dynamics analysis described above was also performed on the binarization resultant interactions. For both the ML and PCL, whole-cell interaction images were drift-corrected and max-projected in FIJI/ImageJ for each of the 12 time points. The custom MATLAB script compared the max-projected time points to determine interaction dynamics. The dynamic interaction index (similar to the M.I. described above) was calculated by dividing the sum of the thresholded extended and retracted pixels by stable pixels between consecutive time points to determine how microglia–Purkinje cell interactions change over time. The interaction coverage index was calculated by dividing the interaction pixels by all pixels in the maximum projection of all time points to determine how much area interactions cover over time.

### 2.7. Fixed Tissue Preparation

After two-photon imaging, between P57 and P65, L7^cre^/Ai9^+/−^/Cx3cr1^G/+^ mice were weighed and euthanized with an overdose of sodium pentobarbital (Euthasol, Virbac) and transcardially perfused with 0.1 M phosphate-buffered saline (PBS, pH 7.4) followed by 4% paraformaldehyde (PFA). Brains were extracted and postfixed in 4% PFA at 4 °C for 72 h, then cryoprotected in 30% sucrose and stored at 4 °C until sectioned. Brains were serially sectioned sagittally on a freezing microtome (Microm; Global Medical Instrumentation, Ramsey, MN, USA) at 50 μm thickness to collect the vermis of the cerebellum and stored at −20 °C in cryoprotectant (25% 0.2 M PB, 25% glycerol, 30% ethylene glycol, 20% ddH_2_O) until mounted. For each animal, three nonadjacent vermal sections around the midline were chosen, mounted on slides, and coverslipped with Prolong Gold mounting media (Molecular Probes, Eugene, OR, USA, Cat# P36934).

### 2.8. Confocal Imaging

For each L7^cre^/Ai9^+/−^/Cx3cr1^G/+^ mouse, three sagittal sections including lobules IV/V of the cerebellar vermis were imaged at the URMC Center for Advanced Light Microscopy and Nanoscopy (CALMN) with a Nikon A1R HD confocal microscope with two excitation wavelengths: 488 (filter cube = 450/50) and 561(filter cube = 525/50) simultaneously, using a 10× (Plan Apo λD 10×, 0.45 NA, Nikon, Melville, NY, USA) or 40× water immersion (Apo LWD 40×/WI λS DIC N2, 1.15 NA, Nikon) objective. Imaging and analyses were performed blind to treatment and sex. Fixed-tissue image analysis was run with Ilastik, NIH ImageJ or FIJI, MATLAB, Cellpose ([36]; version 2.2.3; RRID:SCR_021716), Napari (napari.org; version 0.4.18; RRID:SCR_022765), and RStudio (posit.co; version 2023.09.1+494; RRID:SCR_000432).

### 2.9. Fixed-Tissue Image Analysis

#### 2.9.1. Microglia Density and Spacing

Confocal z-stacks were collected with a 10× objective and z-step of 2.55 μm and stitched together to examine the entire lobule IV/V to assess microglia density and spacing [37,38]. Twenty-five z-slices were max-projected for each section, and ROIs were drawn and separated for the ML and PCL combined, Granule Layer (GL), and White Matter (WM) using a custom FIJI/ImageJ macro. The area of each layer was measured. In the 488 channel, microglia were identified in each layer automatically in Ilastik. Microglia somas were manually traced to train for pixel classification. Appropriate thresholding and size exclusion criteria were chosen during object classification. The object output of microglia somas was binarized and the number and coordinates of somas were analyzed. For microglia density, the number of microglia was divided by the area for the entire lobule IV/V and each of the three layers separately. Microglia spacing and distribution were examined through a nearest-neighbor calculation based on the soma coordinates from the Ilastik output using a custom MATLAB script. The square of the average nearest-neighbor distance multiplied by microglial density was measured to determine the spacing index for the entire lobule IV/V for each section. For both density and spacing, values for all three sections were averaged to determine the final animal value.

#### 2.9.2. Microglia Morphology: Sholl

Confocal z-stacks were collected with a 40× objective and z-step of 0.3 μm to examine microglia arbor complexity for each section with Sholl analysis as described in [34,37,38]. Microglia were excluded if their entire process arbor was not contained in the XY-axes or their entire cell body was not contained in the Z-axis. Fifty-one z-slices were max-projected for each section, the center of each soma was manually selected, and microglia were individually binarized in ImageJ/FIJI. The Sholl analysis plug-in in ImageJ/FIJI was used with concentric rings at 2 μm intervals out to 100 μm from the soma center. The number of intersections across each ring was used to create Sholl curves for the ML, PCL, GL, WM, or all layers combined. Microglia were localized to the ML, PCL, GL, or WM through visual inspection based on Purkinje cell structures. 

#### 2.9.3. Purkinje Cell Linear Frequency

Confocal z-stacks were collected with a 10× objective and z-step of 2.55 μm and stitched together to examine the entire lobule IV/V to assess Purkinje Cell Linear Frequency. In the 561 channel of the max-projected combined ML and PCL layer from Microglia Density, the length of the PCL was determined by drawing and measuring a line across the Purkinje cell somas with the segmented line tool in FIJI/ImageJ. The number of Purkinje cell somas was automatically identified in Cellpose with a custom human-in-the-loop model based on the cyto2 model. Once an appropriate model was trained in Cellpose, batch processing using the custom Cellpose model occurred in Napari to determine the cell counts. The total number of cells for each section was extracted with a custom RStudio code. To determine linear frequency, the number of Purkinje cell somas was divided by the length of the PCL for each section [39], and then the three sections were averaged together to obtain the final animal value. 

#### 2.9.4. Microglia–Purkinje Cell Interactions

Confocal z-stacks were collected with a 40× objective and z-step of 0.3 μm to examine microglia–Purkinje cell interactions in 3D with a colocalization protocol modified from [37]. Fifty-one z-slice substacks (not max-projected) were created in ImageJ/FIJI and the 488 (microglia) and 561 (Purkinje cell) channels were separated. ROIs were drawn and separated for the ML, PCL, and WM using a custom FIJI/ImageJ macro, and layer volumes were measured. Microglia and Purkinje cells were binarized individually in 3D and their pixels were measured for each section. The binarized images were multiplied and the resultant image histogram was used to measure the colocalized microglia–Purkinje cell pixel numbers in 3D for each section. The individual microglia or Purkinje cell pixels were normalized to the volume of the layer to determine the area occupied by each cell type. The microglia–Purkinje cell overlap was normalized to the number of microglia pixels to determine the microglia–Purkinje cell interaction. For all values, the three sections were averaged together to obtain the final animal value. 

### 2.10. Statistics

Statistical tests and graphing were run using GraphPad Prism 10 statistical analysis software (La Jolla, CA, USA; version 10.1.2 (324); RRID:SCR_002798), and normality of distribution was not considered. When sexes were pooled, students’ unpaired *t*-tests were used to compare ethanol- vs. saline-treated mice. To examine any sex differences in ethanol- vs. saline-treated mice, two-way ANOVAs were used with Bonferroni post-hoc comparisons when appropriate. Paired *t*-tests were used to assess layer-specific changes in microglia. Two-way ANOVAs, repeated measures with Bonferroni post-hoc comparisons were used to examine changes over time in ethanol- vs. saline-treated mice. Detailed statistics are provided in the figure legends. Graphs show mean ± SEM.

## 3. Results

### 3.1. Animals

#### 3.1.1. Dosing and Weights 

From postnatal (P) days 4–9, the human third-trimester equivalent, L7^cre^/Ai9^+/−^/Cx3cr1^G/+^ pups were subcutaneously dosed with 5.0 g/kg/day of either ethanol (EtOH) solution or saline, twice a day, 2 h apart (Figure 1A). This is considered a high binge-level amount, producing blood ethanol concentrations (BECs) of ~450 mg/dL in these mice [24]. Every day before the first dose, pups were weighed (Figure 1B). Pups in all groups gained weight throughout the dosing period. There were main effects of age and treatment when sexes were pooled, and from P7–P9 saline-dosed pups weighed significantly more than ethanol-dosed pups (in post hoc testing). Saline-dosed males weighed significantly more than ethanol-dosed males at P28, but not any other ages. Saline- and ethanol-dosed females had no significant differences in weight at any age. 

#### 3.1.2. Quantitative Real-Time PCR Analysis

Because ethanol has been shown to induce cerebellar neuroinflammation in other mouse models of FASD [4,23,40], we examined whether our exposure model caused similar changes. Using a separate cohort of L7^cre^/Ai9^+/−^/Cx3cr1^G/+^ pups, mice were subcutaneously injected with 5.0 g/kg/day of either ethanol solution or saline from P4–9 as before, but were then examined on P10 (24 h after the final dose of ethanol) for neuroinflammatory markers in the cerebellum with rt-PCR (Figure 1C–E). We saw increases in the expression of pro-inflammatory cytokine, IL-1β, in both males and females (Figure 1C), but only a trend towards an increase in TNF-α when sexes were combined (Figure 1D). The pro-inflammatory chemokine, CCL2, increases were restricted to males (Figure 1E). As we observed mild inflammation in the cerebellum immediately after ethanol exposure in our model, we then examined whether this may lead to alterations in microglia and Purkinje cells later in life. 

#### 3.1.3. Surgeries, Imaging, Perfusion

After dosing, animals were aged to adolescence, ~P28, for cranial-window implantation above lobule IV/V of the cerebellar vermis. Mice were aged again to early adulthood, ~P60, for in-vivo imaging and fixed-tissue analysis. At P60, two-photon (2p) imaging was performed to examine cerebellar microglia and Purkinje cells in the Molecular Layer (ML) and Purkinje Cell Layer (PCL) in vivo at high magnification. After imaging, mice were perfused and brains were harvested to examine the entire lobule IV/V. Brains were sectioned and imaged with confocal imaging. As only a subset of cells and layers can be captured with in-vivo 2p imaging, confocal imaging allowed for a larger sample of cells to be analyzed. 

### 3.2. Microglia

#### 3.2.1. Microglia Density and Distribution 

To determine whether developmental ethanol exposure changes cerebellar microglia in adulthood, we first examined microglial density and distribution in fixed sections (Figure 2A), as ethanol-induced decreases in microglial density in the cerebellum have previously been reported [22]. For the entire lobule IV/V, we found no differences in microglia density for either treatment or sex (Figure 2B). Likewise, we found no differences in microglia spacing, which could be indicative of microglial reactivity between ethanol vs. saline or males vs. females (Figure 2C). While ethanol did not appear to have an effect when the lobule was considered as a whole, we divided the lobule into layers to see if there were any layer-specific differences. We examined the Molecular Layer (ML) and Purkinje Cell Layer (PCL) (analyzed together as one layer), the Granule Layer (GL), and the White Matter (WM) (Figure 2D). In the WM, there was a trend towards a higher microglia density in ethanol-dosed animals compared to saline-dosed animals, due to an elevated density in ethanol-dosed females which did not reach significance. There were no differences in any other layer for either treatment or sex, suggesting that developmental ethanol exposure has minimal effect on adult cerebellar microglia density.

#### 3.2.2. Microglia Morphology: Sholl

Because microglial morphology can change to reflect altered microglia function, and changes in microglial morphology after developmental ethanol exposure have been reported [4,22,23], we wanted to quantify microglial morphology throughout lobule IV/V in fixed sections of the adult cerebellum of ethanol- and saline-exposed mice. We used Sholl analysis, as this kind of analysis has previously been used to determine changes in microglial ramification during physiological and pathological events [34,41,42]. In this analysis, concentric circles are drawn at increasing radii from the microglia soma center to the edges of the processes (Figure 3A), and the number of intersections between the processes and each concentric circle is graphed (Figure 3B–E) as a measure of the complexity of the microglial arbor as a function of distance from the soma. While microglial shapes differed by cerebellar layer, with WM microglia appearing to have the smallest arbor diameter, we found no differences due to sex or treatment in microglial Sholl curves when the layers were analyzed together (Figure 3), or singly (Appendix A) using a fully parametric model-based approach [42]. 

This was further supported by a qualitative analysis of microglia imaged in vivo using 2p microscopy. Because of the limited field of view, only one microglia per layer per animal could be analyzed with in-vivo imaging, precluding quantification (Appendix A). Qualitatively, Sholl curves looked similar between ethanol- and saline-exposed males and females, although in ethanol-exposed females, microglia may be slightly less ramified than in saline females in both the ML and PCL (Appendix A). In the ML, male ethanol microglia may be more ramified than in saline microglia (Appendix A). However, this qualitative assessment is likely caused by diverse morphologies of individual microglia in each animal.

#### 3.2.3. Microglia Dynamics

While we observed no differences in microglial density or morphology after developmental ethanol exposure using static analyses in fixed tissue, many microglial roles rely on the ability of their motile processes to explore their surroundings and interact with other cell types [43]. In young adulthood, we used 2p in-vivo imaging to examine cerebellar microglia dynamics in the ML and PCL. For each of our L7^cre^/Ai9^+/−^/Cx3cr1^G/+^ male and female mice, which have fluorescently labeled microglia, we took a 101 μm z-stack including both the ML and PCL every 5 min for an hour, to collect a total of 12 time points for each animal. To examine whether developmental ethanol exposure affected how microglial processes survey the parenchyma over time in adulthood, the motility index (Figure 4A) was analyzed for microglia in the ML (Figure 4B) and PCL (Figure 4C). No differences between ethanol and saline or males and females were found in either layer. Similarly, no changes were found for the surveillance ratio (Figure 4A), a measure of how much parenchyma microglial processes explore in a one-hour period, in either the ML (Figure 4D) or PCL (Figure 4E). 

Because microglia in the ML and the PCL have been reported to have differences in their motility and surveillance, we compared these measures across the two layers in the same animal (Figure 4F–K) [24,28,32]. As expected, PCL microglia were found to be generally more motile than ML microglia in both saline- and ethanol-dosed animals when sexes were combined (Figure 4F) or separated into males (Figure 4G) and females (Figure 4H). Similarly, PCL microglia surveyed significantly more area than ML microglia in both saline- and ethanol-dosed mice when sexes were pooled (Figure 4I). In males, only saline-dosed mice had a significantly higher surveillance ratio in PCL compared to the ML (Figure 4J). There were trends towards higher surveillance in the PCL versus the ML in both saline- and ethanol-dosed females (Figure 4K). This suggests that layer-specific regulation of microglial dynamics is maintained from adolescence to young adulthood and is unaffected by developmental ethanol exposure. 

#### 3.2.4. Microglia Injury Response 

While microglial dynamics were unaffected by developmental ethanol exposure, we wanted to examine whether other microglial functions may be affected. Microglia respond to infection, insult, and injury, and changes in their ability to respond to pathological stimuli have been observed after developmental perturbations, including potentially after ethanol exposure [30,31,44,45]. To determine whether developmental ethanol exposure affects the ability of microglia to respond to focal injury in adulthood, we examined the dynamic chemotaxis of microglial processes to a laser ablation in vivo. A laser ablation was created using a high-intensity brief laser pulse to the brain parenchyma away from visible microglial processes, and the convergence of microglial processes on this ablation was monitored over 60 min (Figure 5A). To quantify the response, the area between the “front” of the microglial processes and injury core was calculated over time as the normalized proximity. Microglial processes converged on the laser ablation over 60 min in all experimental groups, as shown by decreases in the normalized proximity over time (Figure 5B–D). We did not observe differences in the convergence between experimental groups when the sexes were pooled (Figure 5B) or in males (Figure 5C) or females (Figure 5D). While not significant for our analysis, ethanol-dosed females qualitatively appear to converge on the core faster than saline-dosed females (Figure 5D). 

### 3.3. Microglia–Purkinje Cell Interactions 

While our examination of microglia did not show long-term changes in their density, distribution, morphology, dynamics, or injury response following developmental ethanol exposure, it is possible that their interactions with neurons could be altered in a way that is not evident when studying microglia in isolation. Hence, we decided to analyze microglia–Purkinje cell interactions both in vivo using 2p and in fixed sections. 

#### 3.3.1. Purkinje Cell Linear Frequency

Because changes in Purkinje cell numbers could influence the interactions microglia have with these cells, we first examined Purkinje cell linear frequency in lobule IV/V in fixed sections, as ethanol-induced cell death of this population has been reported [46]. Purkinje cell linear frequency was measured by counting the number of somas of the Purkinje cells along a curved line centered in the PCL (Figure 6A). Ethanol significantly decreased the linear frequency by ~8% compared to saline when sexes were combined (Figure 6B) or analyzed separately (Figure 6C). However, post-hoc testing did not show significant changes in either sex, although in females the ~12% reduction approached significance, while the ~4% reduction in males did not (Figure 6C). This suggests that Purkinje cells may be more vulnerable to developmental ethanol exposure than cerebellar microglia, which can lead to long-lasting changes in adulthood for this cell type. 

#### 3.3.2. Microglia–Purkinje Cell Interactions in Fixed Tissue

To determine whether developmental ethanol exposure altered microglia–Purkinje cell interactions, the overlap in these two cell types was measured in fixed sections of lobule IV/V in three dimensions at 40× magnification with confocal imaging. The ML, PCL, and WM were analyzed (Figure 7A). The GL was not analyzed due to the low density of Purkinje cell axons in that layer. First, the number of either microglia or Purkinje cell pixels was divided by the volume of the layer to determine whether ethanol altered the volume occupied by either cell type in adulthood (Figure 7B,C,F,G,J,K). In agreement with our measurements of microglial density and morphology (Figure 2 and Figure 3), microglia pixel numbers were not affected by treatment or sex in any layer (Figure 7B,F,J). Similarly, Purkinje cell volume fraction was unaffected by treatment or sex in the ML (Figure 7C). However, in agreement with Purkinje cell linear density measurements (Figure 6), Purkinje cell volume fraction in the PCL was significantly lower in the ethanol-treated animals compared to saline animals (Figure 7G). This effect was significant in females, but not males. There was a trend of an interaction effect for treatment and sex for the Purkinje cell volume fraction in the WM (Figure 7K). Altogether, these analyses suggest that developmental ethanol exposure does not alter the volume fraction occupied by microglia in any layer of the adult lobule IV/V, but that Purkinje cell volume fraction is decreased in the PCL, reflecting a possible loss of Purkinje cells that is more prominent in females. 

Given the ethanol-driven loss of the Purkinje cell volume fraction, we expected that ethanol would reduce PCL microglia–Purkinje cell physical interactions in females. However, our analysis of overlap between the two cell types found no differences between ethanol- and saline-dosed mice for microglia–Purkinje cell interactions in any layer (Figure 7D,H,L). Microglia are very sparse in the cerebellum, ranging from a volume fraction of ~0.005 in the PCL to ~0.02 in the WM, compared to Purkinje cells, which occupy a consistent volume fraction across layers (~0.25 to 0.30). To account for the different density of microglia in the different layers, we normalized the overlap between microglia–Purkinje cell pixels to the number of microglia pixels (Figure 7E,I,M). While normalized cell–cell interactions were unaltered by treatment or sex in the ML (Figure 7E) or WM (Figure 7M), ethanol-dosed female mice had significantly fewer microglia–Purkinje cell interactions than ethanol-dosed male mice in the PCL (Figure 7I). This suggests that microglia–Purkinje cell interactions in the PCL may be susceptible to ethanol exposure in females, due to the loss of Purkinje cell somas. 

#### 3.3.3. Microglia–Purkinje Cell Interactions In Vivo 

Because microglia are motile cells, we also wanted to determine whether developmental ethanol exposure interfered with the dynamics of the interactions between microglia and Purkinje cells. To do this we used the 2p in-vivo images of the ML and PCL collected in three dimensions over an hour and combined information from the microglia and the Purkinje cell channels to determine the interactions between the two cell types (Figure 8). To determine subcellular organization of the cell–cell interactions, microglia and Purkinje cells were identified on the basis of the whole cell, as well as subdivided into their subcomponents (Appendix A). For both the ML and PCL, the subcomponents for microglia were somas and processes (Appendix A). In the ML, Purkinje cell subcomponents were branch points and non-branch-point dendrites (Appendix A) [24,28]. In the PCL, the only Purkinje cell subcomponent measured was the soma (Appendix A). 

In the ML, the volume fractions for the whole microglia and microglial processes, but not somas, were increased in females compared to males, significantly for whole microglia and with a trend for microglia processes (Appendix A). Post-hoc testing revealed that these differences were due to trends towards a higher microglia volume fraction in ethanol females than ethanol males. Additionally, the volume fraction of the whole Purkinje cell or its subcomponents in the ML was not altered by sex or treatment (Appendix A). We then examined the extent of interactions by normalizing them to the number of microglia pixels and averaging the normalized interactions across the 12 time points imaged in either the ML (Figure 8B and Appendix A) or PCL (Figure 8C and Appendix A). The interaction between microglia and Purkinje cells was measured for whole cells (Figure 8B) and each subcomponent type (microglia process–Purkinje cell dendrite, microglia soma–Purkinje cell dendrite, microglia process–Purkinje cell branch, microglia soma–Purkinje cell branch) (Appendix A–D), then normalized to the number of microglia pixels. For whole microglia–whole Purkinje cell interactions in the ML, there was a trend towards a treatment–sex interaction effect, with ethanol-dosed female mice having more microglia–Purkinje cell interactions than saline-dosed females, although the effect did not reach significance (Figure 8B). Microglia-soma–Purkinje-cell-dendrite interactions had a significant treatment–sex interaction effect (Appendix A). However, no treatment or sex differences were discovered in any other subcomponent interaction in the ML (Appendix A). 

Similar to the ML, in the PCL there was a sex trend toward significance for the volume fraction of the whole microglia (Appendix A)) and microglial processes (Appendix A), but not microglia soma (Appendix A), with females having higher volume fractions than males, an effect which did not reach significance. For Purkinje cell somas, there was a significant treatment–sex interaction effect, although the larger female saline volume fraction compared to the saline males did not reach significance (Appendix A). The interaction between whole microglia and Purkinje cell somas was significantly increased in females compared to males, with trends towards significance in the saline group (Figure 8C). A similar significant effect was found in the microglia process–Purkinje cell soma interactions, with females having more interactions than males, an effect that seemed more prominent in the saline group (Appendix A). No differences were found in the microglia-soma–Purkinje-cell-soma interactions (Appendix A). These results were surprising, as our fixed-tissue interaction analysis (Figure 7I) suggested that ethanol may drive stronger effects in females, leading to a loss of interactions compared to males.

To further examine the dynamic interactions between microglia and Purkinje cells, time points were compared to each other to determine how the interactions evolve over time (Figure 8D–H). A dynamic interaction index was determined in a similar manner to the motility index reported in Figure 4. Adjacent time points representing the binarized whole cell interactions were compared and changes in interactions were quantified (Figure 8D). In the ML, there was a trend towards a treatment–sex interaction effect whereby females trended towards having a lower dynamic interaction index after ethanol treatment compared to saline (Figure 8E). There were no differences in the PCL dynamic interaction index (Figure 8G). To examine how the interactions covered the volume of the parenchyma over time, the sum of the interactions over time was quantified. No differences were found in the ML coverage index (Figure 8F). However, in the PCL, the coverage index was significantly affected by sex, with more coverage in females than in males, an effect that trended towards significance in the saline group (Figure 8H). Overall, in-vivo microglia–Purkinje cell interactions appear to be largely unaffected by developmental ethanol exposure, although sex-specific differences may occur, with female mice having more interactions than male mice in the PCL. 

## 4. Discussion

While the effects of developmental ethanol exposure on microglia dynamics and interactions with neurons have only recently begun to be explored, previous reports have found relatively undisturbed microglial dynamics in the cortex and cerebellum in adolescent mice that were given ethanol in development [24,30,31]. However, whether delayed effects are seen in adulthood has not been determined. Microglia dynamically interact with Purkinje cells in the healthy adult cerebellum [28], suggesting that ongoing interactions between these two cell types may be important to cerebellar physiology and could be impaired by previous ethanol exposure. Here, we examined adult cerebellar microglia and Purkinje cells with in-vivo two-photon imaging, allowing assessment of microglial dynamics, and confocal microscopy in fixed tissue, which allowed us to assay larger samples of cerebellar microglia and neurons than is possible with in-vivo imaging. We focused on young adult animals which were exposed to a binge-level dose of ethanol or saline during the human third-trimester equivalent. While we reported mild inflammation in the cerebellum immediately after ethanol exposure, we found no significant changes in adult microglia density, distribution, morphology, dynamics, or injury response elicited by developmental ethanol treatment. However, we found that ethanol-treated mice had a significantly lower Purkinje cell linear frequency and volume fraction than saline-treated animals in fixed tissue, an effect that was particularly prominent in females. Microglia–Purkinje cell interactions were largely unaltered by developmental ethanol treatment, although we observed sex differences whereby there were fewer cell–cell interactions in female ethanol-dosed mice compared to male ethanol-dosed mice in fixed tissue in the PCL. Conversely, in the in-vivo PCL, there were more interactions between the two cell types in females than males, which was more prominent in saline-dosed animals. Overall, developmental ethanol exposure does not appear to have a large effect on cerebellar microglia and their interactions with Purkinje cells in adult mice, but does affect Purkinje cells. 

Other studies have examined the effects of developmental ethanol exposure on microglia and Purkinje cells, but our study has some unique considerations. Many studies examine rodents immediately after ethanol exposure, while we explore its later-life effects [22,40]. Here, and in previous studies, we have used a subcutaneous injection model [24,30,31], while other research groups utilize other administration methods such as intragastric gavage during similar time frames [22,23]. Additionally, we specifically examined lobule IV/V of the cerebellar vermis, while other studies report the effects on different lobules [4,13,22]. We examined a mouse model of FASD in a specific mouse that allows us to image microglia and Purkinje cells in vivo, while other groups use a different genotype or rats [10,22,23,46]. Our examination of microglia and Purkinje cells in vivo is novel and has not been explored by other groups. Additionally, for our fixed-tissue analyses, our tissue does not need to undergo immunohistochemistry, which many other groups use to visualize microglia and Purkinje cells. However, we use common analyses such as Sholl analysis and Purkinje cell linear frequency [23,39]. These factors are important to consider and may lead to some of the differences between our results and those of previous studies.

We previously examined the effects of developmental ethanol exposure on the adolescent mouse brain [24]. Using the same dosing paradigm, we found that in-vivo microglia dynamics, morphology, and interactions with Purkinje cells in adolescent mice were largely unaltered by a binge-level dose of ethanol from P4–9 [24]. However, we wanted to examine the possibility that differences not apparent in adolescence would be unmasked in adulthood when the cerebellum is fully developed. Our results suggest that microglial dynamics and interactions with Purkinje cells are maintained with age, as we found many of the same results in adulthood as in adolescence. Microglia motility and surveillance, as well as in-vivo morphology, was unaffected by ethanol at both ages. At both ages, microglia motility and surveillance were higher in the PCL compared to the ML, confirming previous results in control adult mice [28]. Microglia–Purkinje cell interactions in vivo were likewise largely unaffected by ethanol exposure at both ages. Some interesting sex differences appeared, with females being primarily affected by developmental ethanol exposure at both ages in vivo. In adolescence, there were trends towards more ML cell–cell interactions in ethanol-treated mice compared to saline-treated mice, with female mice being particularly affected. Similarly in adulthood, for in-vivo ML whole microglia–whole Purkinje cell interactions, there was a trend of increased ethanol female interactions compared to saline females. While there were no differences in adolescent PCL interactions, there were significant sex differences in young adult PCL interactions in vivo. Microglia (whole and process)–Purkinje cell soma interactions were significantly higher in females than males, an effect that appeared largely in the saline group. Microglia–Purkinje cell interactions in vivo appear to have age-dependent sex and layer differences. 

While in-vivo two-photon microscopy allows us to assay dynamic properties of microglia such as their motile processes as they interact with neurons and navigate to sites of injury, this technique is limited by its small field of view, which allows us to examine only a small subset of microglia and Purkinje cells in the ML and PCL of lobule IV/V. To expand our study to include the entire lobule IV/V with all layers, and a larger sample size of cells, we examined fixed tissue with confocal imaging. In the ML and WM, fixed tissue microglia–Purkinje cell interactions were unaffected by treatment or sex. As in-vivo ML interactions were not greatly affected, this was not unexpected. It was surprising that we saw significantly fewer interactions in ethanol-dosed female mice than ethanol-dosed male mice in the fixed-tissue PCL, as there was a main effect of sex in the in-vivo PCL, with female mice having significantly more interactions than male mice overall. These variations may be caused by the differences in limited interactions sampled in vivo or may be due to timescale, with fixed tissue being imaged at a single point in time and in-vivo imaging occurring over an hour. In the case of the fixed-tissue results, the changes in interactions appear to be caused by changes in Purkinje cells, while the changes in vivo may be caused by changes in microglia. 

The lack of large effects of ethanol on microglia–Purkinje cell interactions may be due to the fact that we did not see large ethanol-induced changes in microglial density, morphology, or dynamics. We found young adult microglia density and spacing to be largely unaffected by P4–9 ethanol exposure, with only a trend for ethanol-dosed animals to have a higher density of microglia in the WM than saline-dosed animals. When examining microglia volume fraction in fixed tissue at a higher magnification, no treatment or sex differences were found in any layer. However, when examining microglia in vivo, female mice had an increased volume fraction compared to male mice for whole microglia and microglia processes in the ML and PCL, an effect which appeared more prominent in the ethanol group. These variations are also likely caused by the microglia sampled using the two techniques or the timescale of observation. A previous study found decreased microglia density after a similar developmental ethanol exposure, although exposure was achieved using gavage, and microglia were assessed in lobule IX of the cerebellar vermis at an early postnatal time point [22]. Given that microglia have a particularly robust potential to proliferate throughout life [47], it is possible that early loss of microglia could be compensated by increased proliferation to replace those microglia at later ages. Methodological differences, such as the fact that we assessed density rather than total cell number using stereological methods, could also influence our conclusions. Additionally, our study focused on lobule IV/V of the cerebellar vermis and used subcutaneous injections of ethanol. Differences in ethanol administration method, as well as lobule-specific changes, may cause some of these discrepancies. Various studies have reported altered microglia morphology immediately after developmental ethanol exposure, with microglia becoming more amoeboid or hyper-ramified [4,22,23,37,40]. However, these changes resolve later in life [4], which may explain why we found no differences in adult fixed-tissue microglia morphology in any cerebellar layer due to treatment or sex. Overall, in fixed tissue, microglia appear to be unaltered by ethanol in our study, suggesting that microglia do not undergo long-term changes in density or morphology.

In some instances, developmental insults can result in microglial priming, without overt changes to microglia when assessed at baseline [30,31,44,45]. Priming effects become evident on the presentation of a secondary insult or “second hit” later in life which elicits a larger microglial response. In fact, developmental ethanol appears to exert a mild priming on cortical microglia, whereby these microglia respond more robustly to a laser ablation injury in mice dosed with ethanol in development, although this effect was not statistically significant [30,31]. Additionally, when adult mice were given ethanol, cerebellar microglia responded more robustly towards an injury than controls [32]. Hence, we were interested in determining whether developmental ethanol exposure would also increase the microglial response to focal injury in adulthood. However, we found no differences in adult microglia injury response in the cerebellum, suggesting that any potential microglia priming elicited by ethanol has resolved by this age or that cerebellar microglia are not primed by developmental ethanol exposure. It is also possible that ethanol-induced priming in cerebellar microglia requires a more robust stimulus. An alternate insult in adulthood, such as lipopolysaccharide (LPS), or a second insult in development could be explored in the future to potentially unmask such changes. 

In our previous study using in-vivo imaging in adolescent animals, we were also unable to assay the density of Purkinje cells [24]. The reduction of Purkinje cell linear frequency and volume fraction in the PCL in fixed tissue in adulthood is consistent with previous studies that have found reduced Purkinje cell numbers immediately after ethanol exposure, as well as later in life [4,22,46]. As Purkinje cell linear frequency examines the PCL and there was a decrease of the Purkinje cell volume fraction in the PCL rather than the ML, the Purkinje cell somas appear to be primarily affected by developmental ethanol exposure as assayed in fixed tissue. Purkinje cell soma volume fractions in vivo only had a significant treatment–sex interaction with a trend for higher saline female than saline male. Again, these differences may be due to the difference in the number of Purkinje cells we were able to sample between fixed and in-vivo images. While there was only a trend for a treatment–sex interaction effect for fixed-tissue Purkinje cell axon volume fraction in the WM, we were unable to image the WM in vivo due to depth limitations. The impacts of developmental ethanol exposure on Purkinje cells may be species-specific, as studies in rats have reported severe loss of Purkinje cells after developmental ethanol exposure [10,46]. Bergmann glia somas also reside in the PCL and enwrap Purkinje cells supporting their function; thus, it would be interesting to examine whether they are also altered in our model of developmental ethanol exposure [17,48]. Other studies have found altered Bergmann glia maturation after ethanol exposure [49,50,51]. Along with decreased Purkinje cell numbers, another study found that developmental ethanol exposure led to increased Purkinje cell simple spike firing rates, decreased intrinsic excitability, and altered synaptic connections with parallel fibers, but no changes in climbing fiber synapses [19]. Our work confirms that developmental ethanol exposure affects this cell type and that future studies should further examine the surviving Purkinje cells, as well as other cell types, to see whether cerebellar circuity is further altered. 

Our findings suggesting that microglia are more resistant than Purkinje cells to long-lasting changes after developmental ethanol exposure are consistent with other reports in the literature. While Purkinje cell number has been found to be decreased at P45 and P60 after developmental ethanol exposure, cerebellar microglia morphology was unaltered at P45, suggesting Purkinje cells are particularly vulnerable to long-term changes [4,52]. Additionally, while we found mild increases in pro-inflammatory markers in the cerebellum immediately after developmental ethanol exposure, others have reported that these changes resolve by P45, suggesting that the neuroimmune system may return to homeostasis [23]. While our work suggests that developmental ethanol exposure does not cause prolonged changes in microglia that last until adolescence and young adulthood, animals aged past young adulthood, and especially into old age, may be prone to other deficits. Supporting this, auditory brainstem abnormalities in rats were reported in middle-aged adulthood, but not young adulthood, after ethanol exposure in development [53]. In the hypothalamus, hippocampus, and cortex, developmental ethanol exposure led to increased microglial and immune responses to LPS at P90 [54,55]. Developmental ethanol exposure may increase risk of adult-onset chronic diseases [56]. Therefore, exploring microglia and Purkinje cells in aged mice may yield insight into how these late onset effects develop. 

## 5. Conclusions

Overall, this study suggests that developmental ethanol exposure has little effect on adult cerebellar microglia density, dynamics, morphology, injury response, and interactions with Purkinje cells. However, adult Purkinje cell numbers, particularly in females, are reduced by this binge exposure model of the human third trimester, suggesting that this is a particularly important cell type to study. Future work should examine Purkinje cell function and dysfunction in later adulthood and beyond. 

## Figures and Tables

**Figure 1 cells-13-00386-f001:**
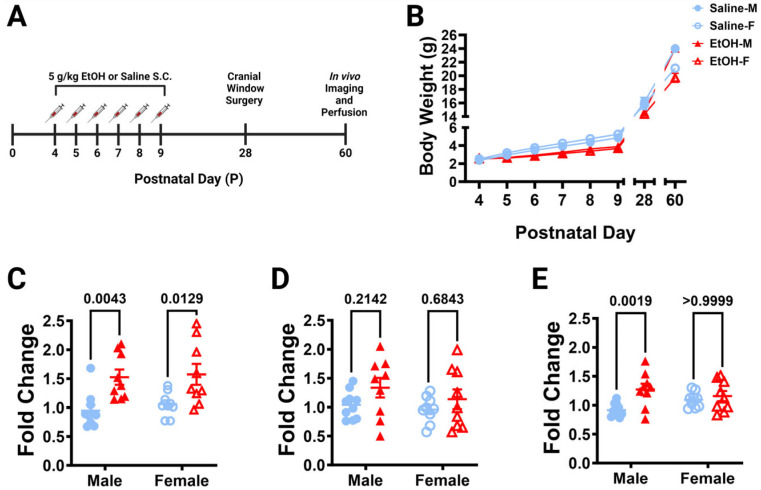
Dosing paradigm, mouse weights, and P10 mRNA expression. (**A**) Timeline of third-trimester equivalent dosing. Pups were subcutaneously given 5.0 g/kg/day of either ethanol solution or saline twice a day, two hours apart. Mice were aged to adolescence (~P28) and cranial windows were implanted over lobule IV/V of the cerebellar vermis. Some animals underwent in-vivo imaging at this age as reported in [24]. They were further aged to young adulthood (~P60) and underwent in-vivo two-photon imaging, followed by fixation. Brains were harvested, sectioned, and imaged on a confocal microscope. (**B**) Mice were weighed every day before dosing (saline 9M; 7F, ethanol 8M; 6F), as well as before surgery and perfusion. When sexes were pooled, there was a main effect of age (F (1.791, 50.16) = 1757, *p* < 0.0001) and main effect of treatment (F (1, 28) = 8.182, *p* = 0.0079) on body weight over time, and a trend towards an interaction between age and treatment (F (7, 196) = 1.980, *p* = 0.0596). Saline-dosed animals gained weight every day (*p* < 0.0001) and ethanol-dosed animals gained weight every day (*p* < 0.05), except for P4–P5 (*p* > 0.99) when sexes were pooled. From P7–P9, saline-dosed pups weighed significantly more than ethanol-dosed pups (*p* < 0.05), but there were no significant differences in weight between saline- and ethanol-dosed animals at any other ages (*p* > 0.05) when sexes were pooled. Saline-dosed males weighed significantly more than ethanol-dosed males at P28 (*p* = 0.0226), but not any other ages (*p* > 0.05). There were no significant differences between saline- and ethanol-dosed female (*p* > 0.05) weights at any age. (**C**–**E**) The effect of P4–P9 developmental ethanol exposure on the P10 (24 h after the final dose of ethanol) cerebellum in a separate cohort of mice (saline 10M; 9F, ethanol 9M; 9F) was examined for IL-1β (**C**), TNF-α (**D**), and CCL2 (**E**) mRNA expression. (**C**) IL-1β expression was significantly increased in ethanol-dosed pups (F (1, 33) = 19.44, *p* = 0.0001), an effect that was significant for both males (*p* = 0.0043) and females (*p* = 0.0129). There was no main effect for sex (F (1, 33) = 0.4066, *p* = 0.5281), and no treatment–sex interaction (F (1, 33) = 0.05975, *p* = 0.8084). (**D**) There was a trend for increased TNF-α expression in ethanol pups (F (1, 33) = 3.409, *p* = 0.0738), but no main effect for sex (F (1, 33) = 1.161, *p* = 0.2892), and no treatment–sex interaction (F (1, 33) = 0.2171, *p* = 0.6443). (**E**) CCL2 expression was significantly increased in ethanol-dosed pups (F (1, 33) = 8.859, *p* = 0.0054) and there was a significant treatment–sex interaction (F (1, 33) = 4.317, *p* = 0.0456), as the effect was significant in males (*p* = 0.0019), but not females (*p* ≥ 0.9999). There was no main effect for sex (F (1, 33) = 0.1921, *p* = 0.6641). (**B**–**E**) Males (M) are shown with closed shapes and females (F) with open shapes. Data are presented as the mean ± SEM. (**B**) Each datapoint represents a treatment group and sex. Two-way ANOVA, repeated measures with Bonferroni post-hoc comparisons. (**C**–**E**) Each datapoint represents an individual animal. Two-way ANOVA with Bonferroni post-hoc comparisons.

**Figure 2 cells-13-00386-f002:**
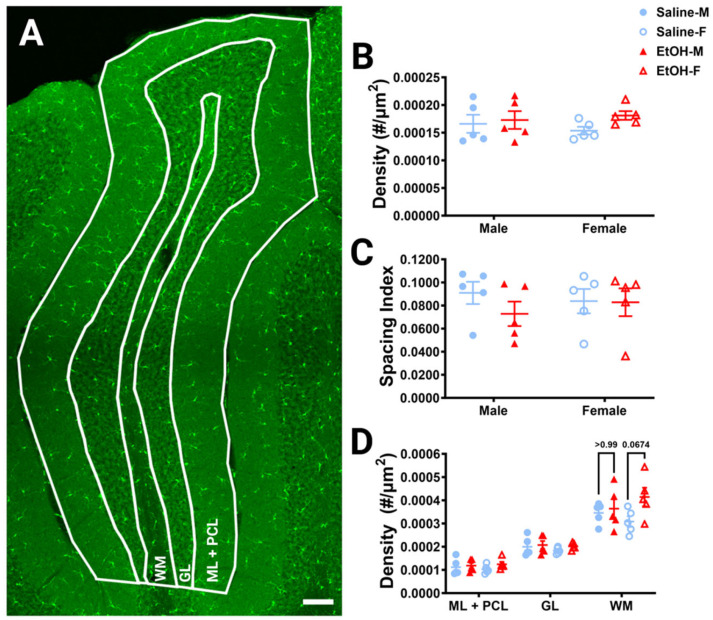
Cerebellar microglia density and distribution in lobule IV/V. (**A**) 10× confocal stitched image of lobule IV/V of the vermis. White lines indicate different layers (Molecular Layer (ML)+ Purkinje Cell Layer (PCL) combined, Granule Layer (GL), White Matter (WM)). (**B**) For the entire lobule IV/V (all layers combined), microglia density (number of microglia/μm^2^) was unchanged between saline-dosed (5M, 5F) and ethanol-dosed (5M, 5F) mice. There was no main effect for treatment (F (1, 16) = 1.836, *p* = 0.1943) or sex (F (1, 16) = 0.02769, *p* = 0.8699), and no interaction between treatment and sex (F (1, 16) = 0.6661, *p* = 0.4264). (**C**) For the entire lobule IV/V (all layers combined), the microglia spacing index ((nearest neighbor)^2^xmicroglia density) was unchanged between saline-dosed (5M, 5F) and ethanol-dosed (5M, 5F) mice. There was no main effect for treatment (F (1, 16) = 0.7948, *p* = 0.3859) or sex (F (1, 16) = 0.01861, *p* = 0.8932), and no interaction between treatment and sex (F (1, 16) = 0.6416, *p* = 0.4349). (**D**) When broken down by layer (ML + PCL, GL, WM), microglia density remained largely unaltered. In the ML and PCL combined, there was no main effect for treatment (F (1, 16) = 1.441, *p* = 0.2474) or sex (F (1, 16) = 0.01629, *p* = 0.9000), and no interaction between treatment and sex (F (1, 16) = 0.3860, *p* = 0.5432). In the GL, there was no main effect for treatment (F (1, 16) = 0.9490, *p* = 0.3445) or sex (F (1, 16) = 0.6814, *p* = 0.4213), and no interaction between treatment and sex (F (1, 16) = 0.1797, *p* = 0.6773). However, in the WM there was a trend towards increased microglia density in ethanol-dosed animals (F (1, 16) = 3.707, *p* = 0.0722), with a trend in females (*p* = 0.0674), but not males (*p* > 0.99). There was no main effect for sex (F (1, 16) = 0.03950, *p* = 0.8450) and no interaction between treatment and sex (F (1, 16) = 1.850, *p* = 0.1927). (**B**–**D**) Each datapoint represents an individual animal. Data are presented as the mean ± SEM. Two-way ANOVA with Bonferroni post-hoc comparisons. Scale bar = 100 μm.

**Figure 3 cells-13-00386-f003:**
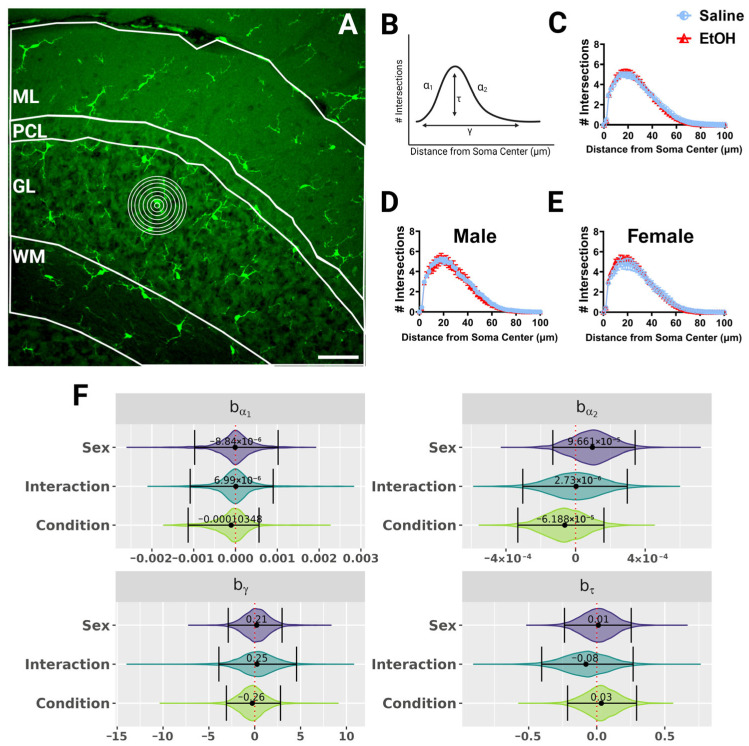
Cerebellar microglia morphology in lobule IV/V in fixed tissue. (**A**) 40× confocal image within lobule IV/V of the vermis. White lines indicate different layers (ML, PCL, GL, WM). Concentric rings were drawn with increasing radii around each microglia (example in GL) to examine process ramification. (**B**) Example Sholl curve graph. X axis corresponds to radii from the soma center (concentric circles in (**A**)) and y axis corresponds to process intersections across the circles. The Sholl curve is labeled with factors that give information about its behavior: before the change-point (α1); after the change-point (α2); branch maximum (eτ); change-point (γ) (from [24]). (**C**–**E**) Sholl curves for saline-dosed (5M, 5F) and ethanol-dosed (5M, 5F) mice for all layers combined when sexes were combined (**C**) or separated into males (**D**) and females (**E**) show no differences in ramification. (**F**) Individual Sholl curves were fit using a hierarchical Bayesian approach to capture variation at each level of the experimental hierarchy. Credible intervals of 95% for effects on each parameter from (**B**) were calculated across treatments and sexes when all layers were combined. (**C**–**E**) Each datapoint represents a treatment group and sex. Data are presented as the mean ± SEM. Scale bar = 50 μm.

**Figure 4 cells-13-00386-f004:**
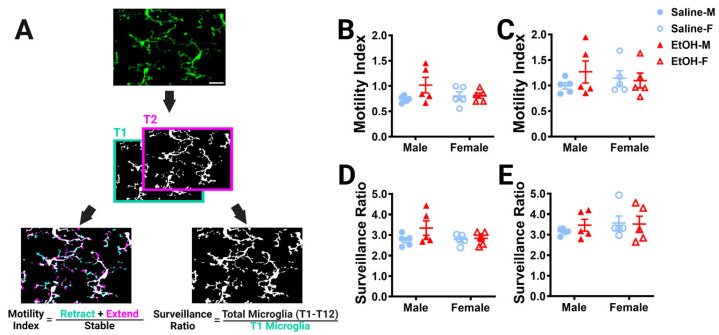
Cerebellar microglia dynamics in vivo. (**A**) Example in-vivo two-photon image of cerebellar microglia. Each time point was binarized (example shows time point 1 (T1) and time point 2 (T2). Each time point was compared to its adjacent time point to find the motility index by taking the sum of the retracted (blue) and extended (pink) pixels divided by the stable (white) pixels. To determine the surveillance ratio, a maximum projection of all 12 time points was created and the total number of microglia pixels in the projection was normalized to the number of microglia pixels in T1 (blue). (**B**,**C**) Microglia motility was unchanged by treatment or sex in both the ML (**B**) and PCL (**C**). There was no interaction between treatment and sex in either the ML ((**B**); F (1, 16) = 2.080, *p* = 0.1685) or PCL (**C**; F (1, 16) = 1.109, *p* = 0.3079). There was no main effect for treatment in either the ML ((**B**); F (1, 16) = 2.343, *p* = 0.1454) or PCL ((**C**); F (1, 16) = 0.5477, *p* = 0.4700). There was no main effect for sex in either the ML ((**B**); F (1, 16) = 0.6480, *p* = 0.4326) or PCL ((**C**); F (1, 16) = 0.003263, *p* = 0.9552). (**D**,**E**) Similarly, microglia surveillance was also unaltered in both the ML (**D**) and PCL (**E**). There was no interaction between treatment and sex in either the ML ((**D**); F (1, 16) = 1.625, *p* = 0.2207) or PCL ((**E**); F (1, 16) = 0.3599, *p* = 0.5570). There was no main effect for treatment in either the ML ((**D**); F (1, 16) = 2.322, *p* = 0.1471) or PCL ((**E**); F (1, 16) = 0.1977, *p* = 0.6625). There was no main effect for sex in either the ML ((**D**); F (1, 16) = 1.282, *p* = 0.2742) or PCL ((**E**); F (1, 16) = 0.6310, *p* = 0.4386). (**F**–**K**) The motility (**F**–**H**) and surveillance (**I**–**K**) of microglia in the ML and PCL of the same animal were compared. PCL microglia were significantly more motile than ML microglia when sexes were pooled ((**F**); saline: *p* = 0.0006, EtOH: *p* = 0.0028). In males (**G**), there was more motility in PCL microglia than ML microglia, an effect which was significant in saline-dosed males (*p* = 0.0269) or exhibited a trend towards significance in ethanol-dosed males (*p* = 0.0582). In females (**H**), PCL microglia were significantly more motile than ML microglia in both saline-dosed (*p* = 0.0200), and ethanol-dosed (*p* = 0.0482) animals. PCL microglia surveyed significantly more area than ML microglia when sexes were pooled ((**I**); saline: *p* = 0.0057, EtOH: *p* = 0.0352). In males (**J**), saline-dosed animals had a higher surveillance ratio in the PCL than the ML (*p* = 0.0291), but there were no differences in ethanol-dosed males (*p* = 0.3767). In females (**K**), there were trends towards higher surveillance in the PCL than ML (saline: *p* = 0.0570, EtOH: *p* = 0.0546). (**B**–**E**) Each datapoint represents an individual animal. Data are presented as the mean ± SEM. Two-way ANOVA with Bonferroni post-hoc comparisons. (**F**–**K**) Each pair of datapoints connected by a line represents an individual animal. Paired *t*-tests. Scale bar = 25 μm.

**Figure 5 cells-13-00386-f005:**
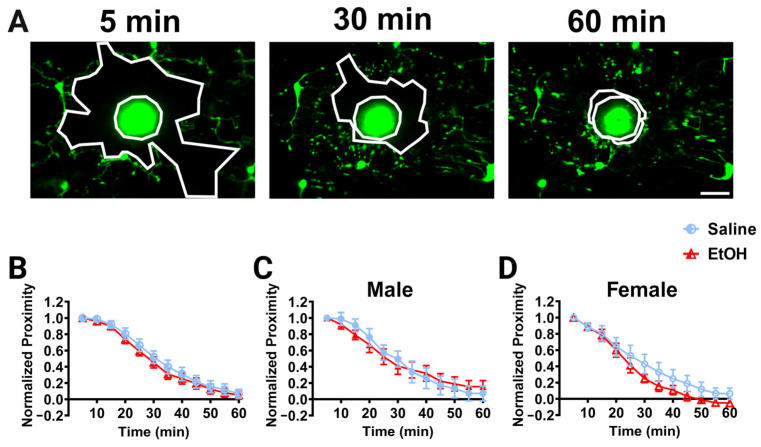
Cerebellar microglia injury response in vivo. (**A**) Example in-vivo two-photon images of cerebellar microglia and the injury core (bright area of autofluorescent debris) over time after a brief laser ablation was inflicted. Outlined in white in each image are the ablation core and the “microglia front,” which decreases over time as microglia converge on the injury. (**B**–**D**) Quantification of microglial convergence on the injury core over 60 min of imaging. No significant differences between ethanol- and saline-dosed animals for convergence were observed when sexes were combined (**B**), or separated into males (**C**) and females (**D**). When sexes were pooled (**B**), there was a main effect of time (F (2.326, 41.87) = 209.2, *p* < 0.0001) on microglia normalized proximity, but no main effect of treatment (F (1, 18) = 0.3516, *p* = 0.5606) and no interaction between time and treatment (F (11, 198) = 0.3193, *p* = 0.9811). In males (**C**), there was a main effect of time (F (11, 88) = 85.68, *p* < 0.0001), but no main effect of treatment (F (1, 8) = 0.001843, *p* = 0.9668) and no interaction between time and treatment (F (11, 88) = 1.104, *p* = 0.3680). In females (**D**), there was a main effect of time (F (2.877, 23.02) = 117.3, *p* < 0.0001), but no main effect of treatment (F (1, 8) = 1.188, *p* = 0.3075) and no interaction between time and treatment (F (11, 88) = 1.031, *p* = 0.4263). (**B**–**D**) Each datapoint represents a treatment group and sex. Data are presented as the mean ± SEM. Two-way ANOVA, repeated measures with Bonferroni post-hoc comparisons. Scale bar = 25 μm.

**Figure 6 cells-13-00386-f006:**
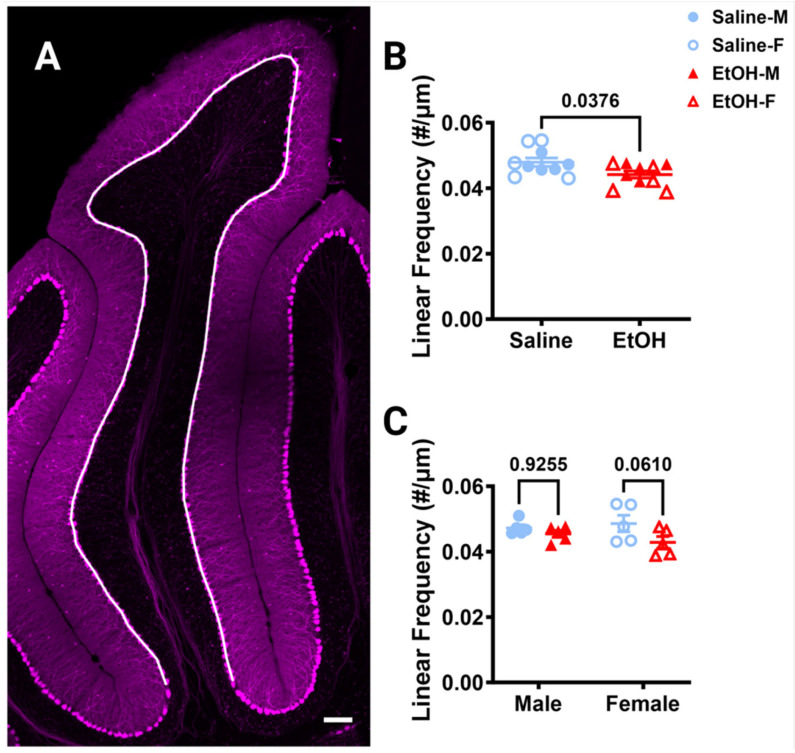
Cerebellar Purkinje cell linear frequency in lobule IV/V. (**A**) 10× confocal stitched image of lobule IV/V of the vermis. White line indicates the length of the PCL. (**B**,**C**) Purkinje cell linear frequency (number of Purkinje cells divided by the length of the PCL) was significantly lower in ethanol-dosed animals when sexes were combined ((**B**), *p* = 0.0376) and when separated by sex ((**C**), F (1, 16) = 4.884, *p* = 0.0420), although this effect seemed to be caused by females where the post-hoc test showed a trend towards significance (*p* = 0.0610), whereas it did not in males (*p* = 0.9255). There was no main effect of sex (F (1, 16) = 0.1277, *p* = 0.7255) and no interaction between treatment and sex (F (1, 16) = 1.313, *p* = 0.2687). (**B**,**C**) Each datapoint represents an individual animal. Data are presented as the mean ± SEM. (**B**) Unpaired *t*-test. (**C**) Two-way ANOVA with Bonferroni post-hoc comparisons. Scale bar = 100 μm.

**Figure 7 cells-13-00386-f007:**
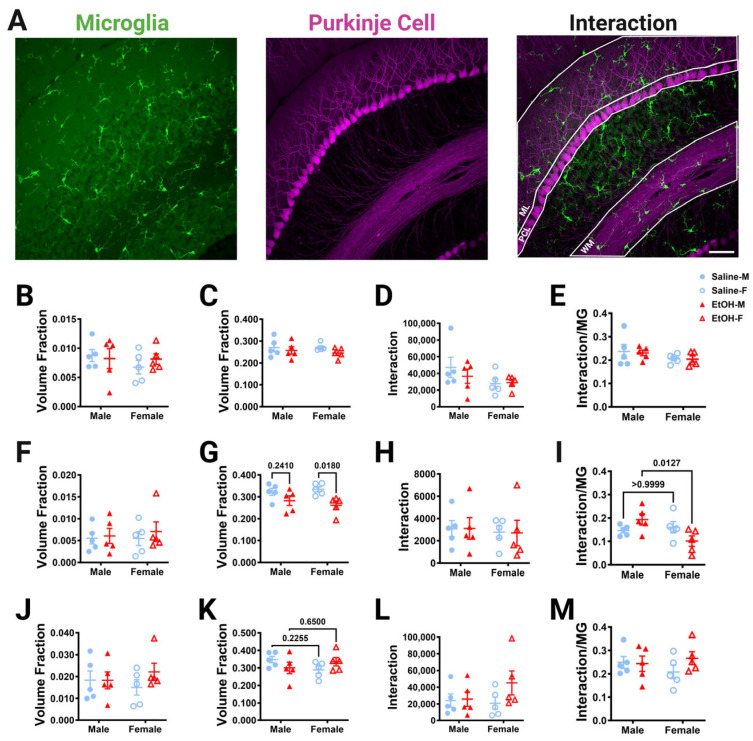
Cerebellar microglia–Purkinje cell interactions in lobule IV/V in fixed tissue. (**A**) 40× confocal images within lobule IV/V of the vermis showing microglia (left), Purkinje cells (middle), and composite (right). White lines indicate different layers that were analyzed (ML (**B**–**E**), PCL (**F**–**I**), WM (**J**–**M**)). (**B**,**C**,**F**,**G**,**J**,**K**) The number of microglia (**B**,**F**,**J**) or Purkinje cell (**C**,**G**,**K**) pixels was divided by the volume of either the ML (**B**,**C**), PCL (**F**,**G**), or WM (**J**,**K**) to obtain the volume fraction. The microglial volume fraction was unaltered in all layers. There was no main effect for treatment in the ML ((**B**); F (1, 16) = 0.1135, *p* = 0.7406), PCL ((**F**); F (1, 16) = 0.3775, *p* = 0.5476), or WM ((**J**); F (1, 16) = 0.8330, *p* = 0.3750). There was no main effect for sex in the ML ((**B**); F (1, 16) = 0.7162, *p* = 0.4099), PCL ((**F**); F (1, 16) = 0.07930, *p* = 0.7819), or WM ((**J**); F (1, 16) = 0.005239, *p* = 0.9432). There was no interaction between treatment and sex in the ML ((**B**); F (1, 16) = 0.5903, *p* = 0.4535), PCL ((**F**); F (1, 16) = 0.09619, *p* = 0.7605), or WM ((**J**); F (1, 16) = 0.8679, *p* = 0.3654). For Purkinje cell volume fraction, there was no main effect for treatment in the ML ((**C**); F (1, 16) = 1.897, *p* = 0.1874) or WM ((**K**); F (1, 16) = 0.001421, *p* = 0.9704). However, there was significantly less Purkinje cell volume fraction in the PCL in ethanol-dosed mice compared to saline-treated mice ((**G**); F (1, 16) = 10.63, *p* = 0.0049), an effect which was significant in females (*p* = 0.0180), but not males (*p* = 0.2410). There was no main effect for sex in the ML ((**C**); F (1, 16) = 0.1437, *p* = 0.7096), PCL ((**G**); F (1, 16) = 0.07170, *p* = 0.7923), or WM ((**K**); F (1, 16) = 0.2196, *p* = 0.6457). There was no interaction between treatment and sex in the ML ((**C**); F (1, 16) = 0.2023, *p* = 0.6589) or PCL ((**G**); F (1, 16) = 0.8862, *p* = 0.3605). However, there was a trend towards a treatment–sex interaction effect for the Purkinje cell volume fraction in the WM ((**K**); F (1, 16) = 3.628, *p* = 0.0750). (**D**,**H**,**L**) The number of pixels representing the overlap between the microglia and Purkinje cell pixels was unaltered by ethanol exposure in every layer. For microglia–Purkinje cell interactions, there was no main effect for treatment in the ML ((**D**); F (1, 16) = 0.3242, *p* = 0.5770), PCL ((**H**); F (1, 16) = 0.0009879, *p* = 0.9753), or WM ((**L**); F (1, 16) = 1.703, *p* = 0.2103). There was no main effect for sex in the ML ((**D**); F (1, 16) = 2.718, *p* = 0.1187), PCL ((**H**); F (1, 16) = 0.1628, *p* = 0.6920), or WM ((**L**); F (1, 16) = 0.6698, *p* = 0.4251). There was no interaction effect between treatment and sex in the ML ((**D**); F (1, 16) = 0.5222, *p* = 0.4804), PCL ((**H**); F (1, 16) = 0.002974, *p* = 0.9572) or WM ((**L**); F (1, 16) = 1.282, *p* = 0.2742). (**E**,**I**,**M**) The number of pixels representing the overlap between the microglia and Purkinje cell pixels was also normalized to the number of the microglia pixels to determine microglia–Purkinje cell interaction. For microglia–Purkinje cell interactions normalized to microglia pixels, there was no main effect for treatment in the ML ((**E**); F (1, 16) = 0.05929, *p* = 0.8107), PCL ((**I**); F (1, 16) = 0.08791, *p* = 0.7707), or WM ((**M**); F (1, 16) = 0.8391, *p* = 0.3732). There was no main effect for sex in the ML ((**E**); F (1, 16) = 2.472, *p* = 0.1354) or WM ((**M**); F (1, 16) = 0.09420, *p* = 0.7629). However, in the PCL (**I**), there was a trend towards a decrease in microglia–Purkinje cell interactions in females compared to males (F (1, 16) = 3.572, *p* = 0.0770) and there was a significant treatment–sex interaction (F (1, 16) = 6.499, *p* = 0.0214), with female ethanol-dosed mice having fewer microglia–Purkinje cell interactions than male ethanol-dosed mice (*p* = 0.0127). There was no interaction effect between treatment and sex in the ML ((**E**); F (1, 16) = 0.01089, *p*= 0.9182) or WM ((**M**); F (1, 16) = 1.229, *p* = 0.2841). (**B**–**M**) Each datapoint represents an individual animal. Data are presented as the mean ± SEM. Two-way ANOVA with Bonferroni post-hoc comparisons. Scale bar = 50 μm.

**Figure 8 cells-13-00386-f008:**
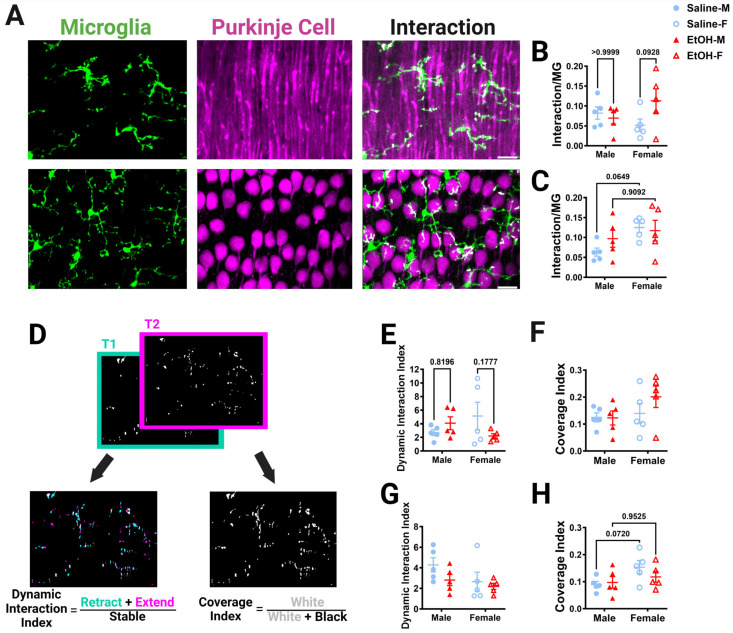
Cerebellar microglia–Purkinje cell interactions in vivo. (**A**) In-vivo two-photon images in the ML (top) and PCL (bottom) showing microglia (left), Purkinje cells (middle), and interactions (right). White overlay indicates microglia–Purkinje cell overlap. (**B**,**C**) The overlap between the whole microglia and whole Purkinje cell pixels was normalized to the number of the microglia pixels to determine microglia–Purkinje cell interaction. For in-vivo whole-microglia–whole-Purkinje cell interactions, there was no main effect for treatment in the ML ((**B**); F (1, 16) = 1.465, *p* = 0.2438) or PCL ((**C**); F (1, 16) = 0.4931, *p* = 0.4926). There was no main effect for sex in the ML ((**B**); F (1, 16) = 0.1041, *p* = 0.7511). In the PCL (**C**), there were significantly more interactions in female mice than in male mice (F (1, 16) = 4.832, *p* = 0.0430), with post-hoc testing showing a trend in saline-dosed mice (*p* = 0.0649). In the ML (**B**), there was a trend towards a significant treatment–sex interaction (F (1, 16) = 3.397, *p* = 0.0839), with ethanol-dosed female mice trending towards having more microglia–Purkinje cell interactions than saline-dosed females (*p* = 0.0928). There was no interaction effect between treatment and sex in the PCL ((**C**); F (1, 16) = 1.242, *p* = 0.2816). (**D**) To determine the dynamic interaction index, the binarized whole-microglia–whole-Purkinje cell overlap (example from ML in (**A**)) in each of the 12 time points was compared to adjacent time points by taking the sum of the retracted (blue) and extended (pink) pixels and dividing by the stable (white) pixels. For the coverage index, the binarized interaction images at all time points were summed (white) and divided by the total pixels (white + black). (**E**,**G**) For the dynamic interaction index, there was no main effect for treatment in either the ML ((**E**); F (1, 16) = 0.4659, *p* = 0.5047) or PCL ((**G**); F (1, 16) = 2.216, *p* = 0.1560). There was no main effect for sex in either the ML ((**E**); F (1, 16) = 0.05820, *p* = 0.8124) or PCL ((**G**); F (1, 16) = 2.947, *p* = 0.1053). In the ML (**E**), there was a trend towards significance for a treatment–sex interaction effect for the dynamic interaction index (F (1, 16) = 3.533, *p* = 0.0785). There was no interaction between treatment and sex in the PCL ((**G**); F (1, 16) = 0.5999, *p* = 0.4499). (**F**,**H**) For the coverage index, there was no main effect for treatment in either the ML ((**F**); F (1, 16) = 0.9049, *p* = 0.3556) or PCL ((**H**); F (1, 16) = 0.4976, *p* = 0.4907). There was no main effect for sex in the ML ((**F**); F (1, 16) = 2.240, *p* = 0.1540). In the PCL (**H**), female mice had a significantly higher coverage index than male mice (F (1, 16) = 4.556, *p* = 0.0486), with post-hoc testing showing a trend in saline-dosed animals (*p* = 0.0720). There was no interaction between treatment and sex in the ML ((**F**); F (1, 16) = 1.033, *p* = 0.3246) or PCL ((**H**); F (1, 16) = 1.216, *p* = 0.2864). (**B**,**C**,**E**–**H**) Each datapoint represents an individual animal. Data are presented as the mean ± SEM. Two-way ANOVA with Bonferroni post-hoc comparisons. Scale bar = 25 μm.

## Data Availability

The raw data supporting the conclusions of this article will be made available by the authors on request.

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
