# Peer review of "Developmental Ethanol Exposure Impacts Purkinje Cells but Not Microglia in the Young Adult Cerebellum"

_cells, 2024, doi:10.3390/cells13050386_

Round 1
Reviewer 1 Report
Comments and Suggestions for Authors
This study comprehensively characterized the impact of developmental ethanol exposure on the cerebellar microglia and Purkinje cells, utilizing a human third trimester binge exposure model in mice. The use of advanced techniques such as two-photon imaging and cranial windows allows for in vivo examination of microglia and Purkinje cells in young adulthood. The study's focus on specific aspects, such as microglia density, morphology, dynamics, injury response, and Purkinje cell linear frequency, enhances the depth of understanding. Notably, the inclusion of sex-specific analysis is a positive aspect of the study, revealing altered microglia-Purkinje cell interactions in females compared to males. The study's contributes valuable insights into the complexities of ethanol's impact on the developing cerebellum. There are only a few minor issues for the authors' consideration.
Lines 21 and 56: Purkinje cells are the sole output of the cerebellar cortex.
Line 115: promoter
Line 136-137: since mice from the same litter were treated with ethanol or saline, these must have been tagged but the procedure is not mentioned.
Line 143: mice solely generated
Line 174: Because thus is an adverb in its own right, the adverbial -ly adds nothing
Line 205: were used
Line 399: please clarify if normality tests were performed prior to choosing parametric tests to analyze the data
Line 424: is it known if this exposure paradigm alters maternal care or causes hypothermia in the pups? What is the behavioral impact of the paradigm (does it induce motor incoordination?)?
Line 941: The Drew and Kane laboratories have demonstrated that a similar exposure paradigm is toxic to cerebellar microglia. A more detailed discussion of the reasons for the discrepancies between those studies and the present study is warranted.
Line 1001: consider that the sensitivity of Purkinje cells to developmental ethanol exposure may be species-specific. Studies with rats have found more severe effects of ethanol.
Author Response
We thank the reviewers for their careful reading of the manuscript and their thoughtful comments. Below we detail the changes made to the manuscript for this resubmission.
Figure and text corrections:
We noticed an error in our reporting of body weights in 3.1.1. Dosing and Weights and Figure 1B. We have fixed these errors editing lines 431-438 of the Results main text, Figure 1B, and lines 449-461 of the Figure 1 legend with the correct information. There was also a small error in Figure 4C which we have updated.
The changes have not changed the main conclusions drawn in these sections and we deeply apologize for these errors.
Reviewer 1
Lines 21 and 56: Purkinje cells are the sole output of the cerebellar cortex.
We have updated these lines per the reviewer’s suggestions.
Line 115: promoter
We have updated this line per the reviewer’s suggestions.
Line 136-137: since mice from the same litter were treated with ethanol or saline, these must have been tagged but the procedure is not mentioned.
We toe-clipped the mice on P4 prior to the first injection for identification. We have included a statement on line 137.
Line 143: mice solely generated
We have updated this line per the reviewer’s suggestions.
Line 174: Because thus is an adverb in its own right, the adverbial -ly adds nothing
We have updated this line per the reviewer’s suggestions.
Line 205: were used
We have updated this line per the reviewer’s suggestions.
Line 399: please clarify if normality tests were performed prior to choosing parametric tests to analyze the data
Many of the data that we present were not normally distributed including data shown in Figures 1B; 2B-D; 4B-C and 4E-K; 5B-D; 7D-E and 7H; 8E; Supp 3D and 3G; Supp 4B-D and Supp 4F. Prior to carrying out the study, we consulted with our biostatistician (Dr. Matthew McCall, who is one of the authors on the paper) and he advised us to not test for normality before using parametric tests. We have added a statement about this on lines 414-415. The reason is well summarized by Lumley et al., 2002 (PMID: 11910059): “Formal statistical tests for Normality are especially undesirable as they will have low power in the small samples where the distribution matters and high power only in large samples where the distribution is unimportant.” For the Sholl curves in Figure 3 and Supplementary Figures 1 and 2, we did not assume normality and instead assumed the Sholl curve process crossings follow a Poisson distribution in the model, which is common for this type of count data.
Line 424: is it known if this exposure paradigm alters maternal care or causes hypothermia in the pups? What is the behavioral impact of the paradigm (does it induce motor incoordination?)?
We were unable to find reports on the impacts of this paradigm on maternal care and if it causes hypothermia in the literature. Our mixed litter design should minimize differences in care between ethanol and saline dosed mice. We also did not notice differences in maternal behavior during dosing. Some ethanol-dosed pups did appear more lethargic following dosing, but were not formally examined for behavior. We have included this information in lines 145-147.
To reduce the chances of hypothermia, pups were placed on a heating pad during dosing when they were removed from the dam. We have added this in lines 139-141.
Line 941: The Drew and Kane laboratories have demonstrated that a similar exposure paradigm is toxic to cerebellar microglia. A more detailed discussion of the reasons for the discrepancies between those studies and the present study is warranted.
We have added a discussion in lines 998-1008 about the possibility that methodological differences such as route of administration (subcutaneous injection vs gavage), examination of different lobules (IV/V vs IX), and assessing density vs. cell number using stereology could affect our results. We also discuss potential microglial proliferation later in life.
We have also added a paragraph to the discussion comparing our methods to methods of other research groups in lines 934-949.
Line 1001: consider that the sensitivity of Purkinje cells to developmental ethanol exposure may be species-specific. Studies with rats have found more severe effects of ethanol.
We have included a statement addressing this in lines 1047-1049 and included appropriate references.
Reviewer 2 Report
Comments and Suggestions for Authors
* This study aims to build upon a previous one on the subject. It proposes that exposure to ethanol during development has minimal impact on various aspects of adult cerebellar microglia. Nonetheless, the number of adult Purkinje cells, especially in females, is diminished by this model of binge exposure resembling the human third trimester, indicating the significance of studying this specific cell type.
* In the last paragraph of the introduction section, you tell us about the results of the study; I would remove it from there. What I would put another information. Given what has been explained in the introduction, what is the question that your study aims to answer?"
* I believe it would be great to include a diagram to better understand Section 2.2.
* So, how many litters did you use? Did all the same litters have the same number of pups?"
* I think that Section 2.6. could be improved. Too concise, please elaborate.
* The results section is written in a convoluted manner, making it difficult to comprehend. I would recommend only including significant values in the figure legends, while presenting the statistical values in tables and explaining them in the body of the text.
* I don´t understand the Individual Sholl curves. Could you briefly explain to me how those graphs are interpreted?
* Can I compare your results with those of other authors? Are these same methods used by other research groups?
Author Response
Reviewer 2
* In the last paragraph of the introduction section, you tell us about the results of the study; I would remove it from there. What I would put another information. Given what has been explained in the introduction, what is the question that your study aims to answer?"
We thank the reviewer for this comment. We expanded this section to make the rationale for the present study clearer in lines 89-97 while keeping a short recap of the results to orient the reader to the study.
* I believe it would be great to include a diagram to better understand Section 2.2.
We have included a reference to Figure 1A on line 139 which presents a diagram of the dosing paradigm.
* So, how many litters did you use? Did all the same litters have the same number of pups?"
Per line 154, 17 litters were used. Pups were not culled so litter size varied between 4-11 pups and we have added this statement to lines 147-148.
* I think that Section 2.6. could be improved. Too concise, please elaborate.
We apologize for the confusion. Each step outlined in Section 2.6 (lines 244-247) is further expanded in subsections 2.6.1-2.6.6. We now include a statement on line 247 stating that more details are provided in the sections below.
* The results section is written in a convoluted manner, making it difficult to comprehend. I would recommend only including significant values in the figure legends, while presenting the statistical values in tables and explaining them in the body of the text.
We appreciate the reviewer’s comments on this. We attempted to reorganize the presentation of the statistics into tables, providing only the significant values in the figure legends, but felt that this attempt had the opposite result making the text cumbersome and more confusing. Thus, we reverted to our previous presentation. If this is a critical comment we are happy to provide the tables but feel that it does not simplify the presentation of the results.
* I don´t understand the Individual Sholl curves. Could you briefly explain to me how those graphs are interpreted?
Sholl analysis examines microglia process ramification. We drew concentric circles at increasing radii from the microglia soma center to the edges of the processes (shown in Figure 3A). The x axis is the Distance (of the rings) from Soma Center in microns. The y axis is the number of times the microglial processes cross the rings for each given radius. We have included a short summary of this analysis in the text on lines 556-559. We used a hierarchical Bayesian approach (Vonkaenel et al., 2023) to statistically analyze these curves between saline and ethanol dosed animals. This analysis is described on lines 580-582 of the Figure 3 legend: “The Sholl curve is labeled with factors that give information about its behavior: before the change-point (α1); after the change-point (α2); branch maximum (eτ); change-point (γ).” The branch maximum is the maximum number of process crossings across all radii. The change-point is the radius where the maximum number of crossings (branch maximum) is observed. There were no effects on any parameter, so the microglia process morphology was not different between conditions.
* Can I compare your results with those of other authors? Are these same methods used by other research groups?
We have included a new paragraph in the Discussion in lines 934-949 to answer this question. We have differences in timeline, administration method, lobule, genotype/model, in vivo imaging, interaction analysis. Our lab performs novel in vivo imaging, which many other research groups do not use. Our examination of microglia-Purkinje cell interactions is novel and there is no literature from other groups to compare to our results. We use fixed tissue analysis techniques commonly used by other groups, such as Sholl analysis and Purkinje cell linear frequency.